# Characterization of Terpenoids from the Ambrosia Beetle Symbiont and Laurel Wilt Pathogen *Harringtonia lauricola*

**DOI:** 10.3390/jof9121175

**Published:** 2023-12-07

**Authors:** Zhiqiang Zhu, Chenjie Yang, Nemat O. Keyhani, Sen Liu, Huili Pu, Peisong Jia, Dongmei Wu, Philip C. Stevenson, G. Mandela Fernández-Grandon, Jieming Pan, Yuxi Chen, Xiayu Guan, Junzhi Qiu

**Affiliations:** 1State Key Laboratory of Ecological Pest Control for Fujian and Taiwan Crops, College of Life Sciences, Fujian Agriculture and Forestry University, Fuzhou 350002, China; sw123zzq@126.com (Z.Z.); cjyang0525@126.com (C.Y.); m17633615410@163.com (S.L.); hdpuhuili@163.com (H.P.); liesleyu@163.com (Y.C.); 2Department of Biological Sciences, University of Illinois, Chicago, IL 60607, USA; keyhani@uic.edu; 3Institute of Plant Protection, Xinjiang Academy of Agricultural Sciences, Urumqi 830091, China; jps-fly@163.com; 4Biotechnology Research Institute, Xinjiang Academy of Agricultural and Reclamation Sciences, Shihezi 832061, China; wdm0999123@sina.com; 5Natural Resources Institute, University of Greenwich, Chatham Maritime, Kent ME4 4TB, UK; p.c.stevenson@greenwich.ac.uk (P.C.S.); m.fernandez-grandon@greenwich.ac.uk (G.M.F.-G.); 6College of Biology & Pharmacy, Yulin Normal University, Yulin 537000, China; jiemingpan@163.com; 7College of Horticulture, Fujian Agriculture and Forestry University, Fuzhou 350002, China

**Keywords:** Ophiostomales, *Harringtonia lauricola*, laurel wilt, secondary metabolite, terpenoid, antimicrobial, antiproliferative, free radical scavenging

## Abstract

Little is known concerning terpenoids produced by members of the fungal order Ophiostomales, with the member *Harringtonia lauricola* having the unique lifestyle of being a beetle symbiont but potentially devastating tree pathogen. Nine known terpenoids, including six labdane diterpenoids (**1**–**6**) and three hopane triterpenes (**7**–**9**), were isolated from *H. lauricola* ethyl acetate (EtOAc) extracts for the first time. All compounds were tested for various in vitro bioactivities. Six compounds, **2**, **4**, **5**, **6**, **7**, and **9**, are described functionally. Compounds **2**, **4**, **5**, and **9** expressed potent antiproliferative activity against the MCF-7, HepG2 and A549 cancer cell lines, with half-maximal inhibitory concentrations (IC_50_s) ~12.54–26.06 μM. Antimicrobial activity bioassays revealed that compounds **4**, **5**, and **9** exhibited substantial effects against Gram-negative bacteria (*Escherichia coli* and *Ralstonia solanacearum*) with minimum inhibitory concentration (MIC) values between 3.13 and 12.50 μg/mL. Little activity was seen towards Gram-positive bacteria for any of the compounds, whereas compounds **2**, **4**, **7**, and **9** expressed antifungal activities (*Fusarium oxysporum*) with MIC values ranging from 6.25 to 25.00 μg/mL. Compounds **4**, **5**, and **9** also displayed free radical scavenging abilities towards 2,2-diphenyl-1-picrylhydrazyl (DPPH) and superoxide (O^2−^), with IC_50_ values of compounds **2**, **4**, and **6** ~3.45–14.04 μg/mL and 22.87–53.31 μg/mL towards DPPH and O^2−^, respectively. These data provide an insight into the biopharmaceutical potential of terpenoids from this group of fungal insect symbionts and plant pathogens.

## 1. Introduction

Fungi are well known to be rich in compounds termed secondary metabolites that display an astonishingly diverse array of biological and biopharmaceutical properties [1,2]. These include varied classes of compounds ranging from those with potential human health relevance, e.g., antimicrobial, anti-cancerous, and immune-modulatory compounds, to those exploitable in industries ranging from food and agriculture, bioremediation, and even cosmetics [3,4,5]. However, it is estimated that there are more than three million fungi in nature, of which humans have discovered less than 8% [6,7], indicating a rich unexplored diversity of organisms and their bioactive compounds awaiting discovery. Terpenoids (isoprenoids derived from five carbon isoprene units) represent a heterogeneous naturally occurring class of compounds most widely studied in plants, where some have been shown to act as phytohormones, antioxidants, in defense and/or community interactions (e.g., attraction of (beneficial) organisms) [8,9]. In addition, bioprospecting of terpenoids for a wide range of applications is a very active field, with significant efforts expended in applications and platforms for terpenoid production [10,11,12,13]. Terpenoids have been characterized from numerous fungal sources, including a range of Ascomycetes and Basidiomycetes, especially endophytic fungi [14,15,16,17]. However, little is known concerning terpenoids from the Ophiostomales fungal order that includes unique plant pathogens and insect symbionts, particularly with respect to bioactivities with potential biopharmaceutical applications.

*Harringtonia lauricola* (formerly *Raffaelea lauricola*, Ascomycota, Ophiostomales, Ophiostomataceae) is an invasive (to the Southeastern United States) beetle-borne/vectored plant pathogenic fungus affecting members of the Lauraceae family, responsible for laurel wilt disease [18]. The original vector for the fungus was the (invasive to the US) red bay ambrosia beetle, *Xyleborus glabratus*, with the insect and its fungal symbiont (*H. lauricola*) likely introduced from unprocessed wood during transport from Asia to the Eastern United States around the turn of the 21st century [19,20]. The fungus is stored in specialized beetle structures termed mycangia [21], and when released by the beetle into trees during gallery excavation, the fungus can attack mature and otherwise healthy hosts. *H. lauricola* has led to the death of over 300 million trees in the Southeastern United States since its introduction, being responsible for eliminating >85% of red bay trees in endemic regions [22]. Most fungal symbiotic partners of these beetles, however, do not cause significant damage to trees, suggesting unique aspects of *H. lauricola* and its interaction with the tree host that results in disease.

Significant aspects of the biology of *H. lauricola* remain to be characterized. Recent examinations of various physiological growth parameters of *H. lauricola* revealed the potential for cold adaptation (optimal growth temp 15–26 °C) and pH sensitivity (reduced growth at pH > 8.0), as well as sensitivity to a range of fungicides, including various conazoles, prochloraz, dithiocarbamates, and zinc-based fungicides [23]. In addition, growth substrate profiling revealed broad utilization of sulfur- and phosphate-containing compounds, comparatively restricted carbon substrate utilization, and rescue of pH and osmotic sensitivities by specific compounds (e.g., amino acids) [24]. Chemotyping of volatile organic compounds from *H. lauricola* identified VOC dynamics potentially linked to the response of host trees [25]. These included a suite of alcohols, pentanes, hexenes, and heptanes, as well as monoterpenes and terpenes; however, no exact determinations of the latter compounds were made. Thus, overall, there remains limited information on the range and/or types of terpenoid secondary metabolites derived from *H. lauricola*, although genomic analyses indicate a large repertoire of putative secondary metabolite biosynthetic gene clusters [26,27,28]. Our objectives were to (1) identify suitable growth and purification conditions for the isolation of terpenes from *H. lauricola* and (2) examine the biopharmaceutical potential of isolated compounds with respect to antibacterial, antifungal, antioxidant, and antiproliferative activities. We show that when grown in a medium of brown rice coupled with Lauraceae species sawdust, a suite of *H. lauricola*-derived terpenoids could be isolated, including labdane diterpenoids and hopane triterpenes. Using a series of cancer cell lines, target bacterial and fungal species, and reactive oxygen species (ROS), the antiproliferative, antibacterial, and antioxidant activities of these compounds were subsequently tested in vitro. These data provide a new window into the secondary metabolite repertoire of *H. lauricola* that might play functional roles in the unique adaptations of this fungus, including plant pathogenicity and beetle mutualism, as well as providing lead compounds for bioprospecting.

## 2. Materials and Methods

### 2.1. General Experimental Procedures

Genomic DNA was extracted using a fungal DNA mini kit from Omega Bio-tek company (Omega, Guangzhou, China). Polymerase chain reaction (PCR) experiments were performed using a Bio-RAD T100™ Thermal Cycler Endpoint PCR (Bio-RAD, Hercules, CA, USA). One-dimensional (^1^H and ^13^C) and two-dimensional (HSQC, HMBC and ^1^H-^1^H COSY) nuclear magnetic resonance (NMR) spectra were recorded using a Varian Unity BRUKER 600 at 600 MHz (^1^H) and 150 MHz (^13^C) (Bruker, Fällanden, Switzerland). Chemical shifts were expressed by *δ* (ppm) relative to deuterated chloroform (CDCl_3_, *δ*_C_: 77.2, *δ*_H_: 7.27, Macklin, Shanghai, China). The molecular weights of compounds were determined using electrospray ionization mass spectrometry (ESI-MS, Agilent 6500, Santa Clara, CA, USA). Spectral and optical density measurements were taken using a Thermo Fisher Scientific Multiskan SkyHigh (Thermo Scientific, Waltham, MA, USA). For compound purification, Sephadex LH-20 and RP-18 columns (C_18_, 40–60 μm) were purchased from Shanghai Yingxin Laboratory (Shanghai, China). Both silica gel (200–300 mesh, for column chromatography) and silica gel GF254 plates (for preparative TLC analyses) were purchased from Qingdao Marine Chemical Factory (Qingdao, China). Chloroform, methanol, petroleum ether, ethyl acetate, acetone, and dichloromethane used in column chromatography were purchased from Shanghai Sinophape Chemical Reagent Co., Ltd. (Shanghai, China). High-performance liquid chromatography (HPLC) was performed using an Alltech 426 series pumping system equipped with an Alltech UV-201 detector (Alltech, Chicago, IL, USA). Chromatographic-grade methanol and acetonitrile were purchased from Merck (Darmstadt, Germany).

### 2.2. Identification of Fungal Species

The fungal strain RL2022 was initially isolated from ambrosia beetles collected in Fuzhou National Forest Park in Fujian Province, China on 3 July 2022, deposited in the China General Microbiological Culture Collection Center (CGMCC), and identified as *Harringtonia* sp. For morphological examination, the strain was inoculated in the middle of PDA plates, and sterilized cover glasses were inserted at a 45° angle, 1 cm away from the colony inoculation site. Plates were incubated in a biochemical incubator at 26 °C over a 7 d time course. After the slide was covered with mycelium, a drop of cotton blue dye was added, and the cover glass examined for microscopic structures of conidiogenous cells, hyphae, and conidia using a Nikon Ni-U upright microscope (Nikon, Tokyo, Japan).

Genomic DNA was isolated using the Omega E.Z.N.A.^®^ Fungal DNAKit (Norcross, GA, USA). The genomic DNA was used as the template for obtaining ITS (1172 bp), LSU (917 bp) and *β*-tubulin (522 bp) nucleotide sequences by PCR using primer pairs ITS1f/ITS4, LROR/LR5, and T10/Bt2b [29], respectively (Appendix A). PCR fragments were sequenced by Sangon Biotech company (Sangon Biotech, Shanghai, China). Phylogenetic analyses were performed using maximum likelihood (ML) in RAxML 8.2.10, GTRGAMMA model, bootstrap = 1000) [30]. The species and GenBank information used in this study are listed in Appendix A. The fungal strain was stored at the China General Microbiological Culture Collection Center (CGMCC3.24979, Institute of Microbiology, Chinese Academy of Sciences).

### 2.3. Small-Scale Fermentation of Target Strains Based on OSMAC Strategy

#### 2.3.1. Small-Scale Fermentation

Using the One Strain–Many Compounds (OSMAC) strategy, three different media compositions were selected for small-scale fermentation of *H. lauricola* [31,32,33,34] as follows: potato dextrose broth (PDB, potato 200 g/L, glucose 20 g/L, pH 7.0), rice + glucose medium (RGM, rice 700 g/L, glucose 20 g/L, pH 7.0), and rice mixed with *Cinnamomum camphora* sawdust (RSM, rice 700 g/L, *C. camphora* sawdust 50 g/L, glucose 20 g/L, pH 7.0). Uninoculated RSM media (no fungus) was used as the control.

#### 2.3.2. HPLC Analysis

After fermentation, crude extracts (CEs) were prepared from mycelia harvested on filter paper and extracted with ethyl acetate. Samples (500 g wet weight) were extracted 3 times (1L EtOAc each time), with each extraction performed at 30 °C for 60 min and using ultrasonic-assisted wall breaking treatment, and the supernatant was collected and then dried using a rotor evaporator. The CE was then weighed and analyzed by YMC C_18_ HPLC column chromatography (5 μm, 10 × 250 mm). The CE was first dissolved in chromatography-grade methanol to prepare a clarified solution at a concentration of 1 mg/mL. The injection volume was 10 μL, the detection wavelength was 254 nm, and the column temperature was set to 24 °C. The mobile phase was a chromatography-grade methanol (mobile phase A)–water (0.1% glacial acetic acid, mobile phase B) system. The gradient elution conditions of the mobile phase were as follows: 5–100% mobile phase A, 0–60 min and then 100% mobile phase A, 60–70 min at a flow rate of 1 mL/min. Based on the results of HPLC and yield analyses, the conditions for larger-scale (*H. lauricola* metabolite) isolation were determined.

### 2.4. Large-Scale Fermentation

According to the screening results of different media, the best culture medium was determined to be RSM media, which was subsequently used for large-scale fermentation. For large-scale fermentation, 120 flasks (of 15 mL PDB each) were inoculated with 15% mycelium prepared as follows: *H. lauricola* was grown on PDA plates at 26 °C for 5 days. The cultured fungal plates were then divided into 0.5 cm × 0.5 cm fungal agar blocks and then used to inoculate the seed flasks of PDB media. The flasks were subsequently cultured at 26 °C with aeration (160 r/min) for 7 d. Finally, the 15 mL/flask of seed liquid was inoculated into RSM media and grown at 26 °C for 28 d with aeration. After the fermentation, 8 kg (wet weight) of fungal mycelia were harvested by filtration and subsequently extracted using twice the volume of ethyl acetate. The sample was extracted 3 times (72 L total volume of EtOAc), with each extraction performed at 30 °C for 60 min and using ultrasonic-assisted wall breaking treatment. Supernatants were collected by filtration and the three extracts were combined and concentrated using a rotary evaporator to obtain the crude extract (60.0 g).

### 2.5. Purification of H. lauricola Metabolites

All reagents used in the metabolite purification process were of analytical grade. The air-dried extract of *H. lauricola* was suspended in CDCl_3_ (200 mg/mL), and (30 mL) was mixed with C_18_-reversed-phase silica gel, which vaporized the CDCl_3_ to obtain a dry powder. The powder was loaded on the top of a glass column filled with C_18_-reversed-phase silica gel and the sample was then subjected to RP-18 column chromatography and eluted stepwise with a mixture of H_2_O-MeOH [70:30 (3.0 L), 50:50 (4.0 L), 30:70 (4.0 L), 10:90 (4.0 L), 0:100 (6.0 L), *v*/*v*] to obtain five fractions (fractions 1–5). Among them, fraction 2 (3.5 g) was further separated using silica gel column chromatography eluted with petroleum ether and acetone 30:1 to 9:1 to obtain fractions 2.1 (200.0 mg) and 2.2 (150.0 mg), yielding compound **1** (9.3 mg) and compound **6** (11.0 mg). Fraction 2.1 (200.0 mg) was chromatographed on a silica gel column eluted with petroleum ether and EtOAc (9:1, *v*/*v*) to gain compounds **2** (19.0 mg) and **3** (15.0 mg). Fraction 2.2 (150.0 mg) was subjected to Sephadex LH-20 column chromatography and fractions eluted with CDCl_3_-MeOH (1:1, *v*/*v*) to yield compounds **4** (9.0 mg) and **5** (11.0 mg). Fraction 3 (1.0 g) was subjected to silica gel column chromatography and eluted stepwise with 100% petroleum ether to 20% petroleum ether and 80% dichloromethane (100:0–20:80, *v*/*v*) to obtain compounds **7** (58.0 mg), **8** (70.0 mg), and **9** (134.5 mg).

### 2.6. Antiproliferation Assays

The antiproliferative activities of the nine purified *H. lauricola* compounds were tested at five different concentrations (0.1, 1,10, 100, and 200 μM) on a series of cancer cell lines, including human breast carcinoma (MCF-7), hepatocellular carcinoma (HepG2), and lung carcinoma (A549). The cell lines (1 × 10^4^ cell/well) were incubated in 96-well plates at 37 °C for 48 h. Cell viability was measured using an MTT assay as described previously [35], with DMSO used as the negative control and cisplatin as a positive control. The experiment was repeated three times with three technical replicates each. All samples were analyzed by measurement at 570 nm absorbance, and the mean concentration for 50% inhibition (IC_50_, μΜ) was calculated from the determined inhibition data:Inhibition (%) = (A_0_ − A_1_)/(A_0_ − A_2_) × 100%
where A_0_, A_1_, and A_2_ stand for the absorbance control group, experiment group, and blank group, respectively.

### 2.7. Antimicrobial Activity

The antimicrobial activities of compounds **1**–**9** were measured using the Clinical and Laboratory Standards Institute (CLSI) method. The test experimental target strains included the plant pathogenic fungus, *Fusarium oxysporum*; two Gram-negative bacteria, *Escherichia coli* and *Ralstonia solanacearum*; and two Gram-positive bacteria, *Bacillus subtilis* and *Staphylococcus aureus*. The antimicrobial activity was preliminarily determined by the agar filter-paper diffusion method, and the compounds were diluted to 200, 100, 50, and 25 μg/mL as the test sample solution. The activated bacteria were evenly coated on LB medium plate with sterile coating stick on the ultra-clean table, and 0.5 cm × 0.5 cm fungal agar blocks were placed in the center of the PDA plate. Filter paper (6 mm) attached to different concentrations of sample solution was then placed on the plate, incubated in the incubator for 3–5 d (37 °C for bacteria, and 28 °C for fungus), followed by the observation of antibacterial effect. Further investigation of compound concentration and inhibition activity was performed as follows: The concentration of the microbial suspensions used in bioassays was 1 × 10^5^ CFU. The nine compounds and positive controls (ciprofloxacin and streptomycin, antifungal and antibacterial, respectively) were mixed with test microbial suspensions in 96 well plates and incubated in the dark at 37 °C (bacteria) for 24 h or at 28 °C (fungus) for 5 d. Each sample was repeated three times and the absorbance value at 600 nm was measured with a microplate reader. The antimicrobial activity was expressed as MIC (minimal inhibitory concentration, μg/mL) [35].

### 2.8. Antioxidant Activity

#### 2.8.1. Determination of DPPH Radical Scavenging Activity

The DPPH free radical scavenging activities of the nine compounds were measured as reported previously [36]. Assay mixtures of 100 μL of each compound were adjusted to various concentrations (0–100 μg/mL) and then mixed with 150 μL of DPPH-methanol solution and shaken to homogeneity and then left to stand in the dark for 30 min. The color change of DPPH was measured at 517 nm. V_C_ (vitamin C) and BHT (butylated hydroxytoluene) were used as positive controls.
DPPH radical scavenging activity (%) = (A_0_ − A_1_)/A_0_ × 100%
where A_0_ is the blank absorbance group, and A_1_ is the experiment group.

#### 2.8.2. Determination of Superoxide Anion Radical (O^2−^) Scavenging Activity

Superoxide anion radical scavenging activity was measured using a superoxide radical scavenging kit according to the manufacturer’s protocols (Nanjing Jiancheng Institute of Bioengineering, Nanjing, China) which utilized the reaction system of xanthine and xanthine oxidase to produce superoxide free radical (O^2−^). V_C_ and BHT were used as positive controls, and samples were measured at 550 nm.
Superoxide radical-scavenging activity (%) = (A_0_ − A_1_)/A_0_ × 100%
where A_0_ is the blank absorbance group and A_1_ is the experiment group.

### 2.9. Statistical Analyses

All results are shown as the mean ± standard deviation (SD) using three independent readings. SPSS software (version 22.0, IBM Corp., Bethesda, MD, USA) was used for statistical analyses. Duncan’s multiple comparison test was employed to evaluate the significance of differences between means. GraphPad Prism software (version 8.0.2 GraphPad Software Inc., San Diego, CA, USA) was used to complete the statistical assessment.

## 3. Results

### 3.1. Fungal Species Identification

#### 3.1.1. Morphological Description

After the cultivation of strain RL2022 (CGMCC accession # CGMCC3.24979) on potato dextrose agar (PDA), fungal colonies appeared initially wet, with submerged hyphae that were transparent to light grey and smooth, with further growth aerial mycelia developed that were off-white (Figure 1A,B). After 10 days of cultivation, the fungus quickly occupied the entire plate, with a colony diameter of ~6.2 cm. Conidiophores were micronematous and the conidia were solitary in clumps, hyaline, smooth, obovoid, rounded apex, and usually tapering toward base, with typical dimensions of 3.5–13 × 3–4.5 μm (Figure 1C–H), similar to what has been reported for this species [19,37].

#### 3.1.2. Phylogenetic Analyses

Strain CGMCC3.24979 was originally collected from ambrosia beetles and putatively identified as *Harringtonia* sp. To confirm the identity of the strain via molecular characterization, ITS, LSU, and *β*-tubulin sequences (GenBank: OP893642, OP880432, and OP935988) were obtained and combined with those related to other species data from NCBI for multi-locus analyses and coupled to morphological characterization (Figure 2). Phylogenetic analysis was performed using maximum likelihood (ML) in RAxML. The species and GenBank information used in this study are given in Appendix A. The combined gene length was 2609 bp. Isolate CGMCC3.24979 clustered with the other *Harringtonia lauricola* isolates in one branch and nested in it (Figure 2).

### 3.2. Optimization of Growth Conditions

Three different media compositions, PDB, RGM (rice + glucose), and RSM (rice + sawdust), were used to determine conditions for large-scale fermentation and metabolite isolation as detailed in the Methods section. Crude extracts had weights of 144.6 mg (PDB), 249.8 mg (RGM), and 288.3 mg (RSM). In order to exclude the interference of medium raw materials (rice, glucose, and Lauraceae sawdust), RSM medium (no fungus) with the same treatment was set as a control, and the crude extract was obtained according to the same method. For the RSM media, degradation of the insoluble sawdust was seen mostly on the surface of the substrate with circular holes formed by the mycelia evident, and a strong smell of alcohol was noted.

The extracts derived from the three media and controls were analyzed by HPLC as detailed in the Methods section (Figure 3). These data indicated that secondary metabolite abundance (yield) and diversity were highest in the RSM media, and this media was the selection for large-scale fermentation and metabolite isolation as detailed in the Methods section.

### 3.3. Terpenoid Extraction

A large-scale fermentation protocol was used, growing *H. lauricola* in RSM, after which fungal cells were extracted with ethyl acetate, followed by an initial separation by silica gel column chromatography, with further purification using silica gel, reverse-phase C_18_, and Sephadex LH-20 columns as detailed in the Methods section. In total, nine compounds were purified and subjected to 1D and 2D-NMR spectral analyses for structural identification (compounds **1**–**9**, Figure 4). The results of HPLC analyses further verified that these compounds were extracted from *H. lauricola* secondary metabolites rather than medium components (Figure 3C). These compounds were identified as: manool (**1**) [38], 18-hydroxy-7-oxolabda-8(9),13(*E*)-dien-15-oic acid (**2**) [39], 7-oxolabda-8(9),13(*Z*)-diene-15,18-dioic acid (**3**) [40], 3*β*-hydroxy-8(17),13*E*-labdadien-15-oic acid (**4**) [41], enantio-labda-8(20),13(*E*)-dien-15,18-dioic acid (**5**) [42], labd-14-en-19-al,8,13-epoxy (**6**) [43], 15*α*-hydroxyhop-17(21)-ene (**7**) [44], 15*α*-hydroxy-21*α*-*H*-hopane (**8**) [45], and 15*α*,22-dihydroxyhopane (**9**) [46]. Spectra and spectrum data are shown in Appendix A. These nine known terpenoids, including six labdane diterpenoids (**1**–**6**) and three hopane triterpenes (**7**–**9**), were isolated from *H. lauricola* ethyl-acetate (EtOAc) extracts, and even more broadly from Ophiostomatales, for the first time.

### 3.4. Antiproliferation Bioassays

In order to determine the antiproliferative (anti-cancerous) activity of the *H. lauricola* compounds, cytotoxicity bioassays were performed using a variety of human tumor cell lines as detailed in the Materials and Methods section. Cell lines used in the bioassays included human breast carcinoma (MCF-7), human hepatic carcinoma (HepG2), and human lung cancer (A549), with antiproliferative effects compared with the anti-cancer drug cisplatin. Compounds **2**, **4**, **5**, **6**, **7**, and **9** exhibited specific (growth) inhibitory effects against MCF-7, HepG2, and A549 tumor cell lines (Table 1). The IC_50_ values of compounds **2**, **4**, **5**, and **9** against the three tumor cells ranged from 12.54 to 26.06 μM. Interestingly, compound **9** only showed inhibitory effects versus the HepG2 cell line (IC_50_ = 18.81 ± 1.25 μM), whereas compound **6** was active against all three tumor cell lines but inhibitory effects (IC_50_) occurred from 27.31 to 38.46 μM. Compound **7** showed weak antitumor activity with an IC_50_ of 36.52–48.61 μM against the various cell lines.

### 3.5. Antimicrobial Activity

The antibacterial and antifungal activities of the purified *H. lauricola* terpenoids were investigated using three different bacteria and a phytopathogenic fungus. Bacterial targets included Gram-negative (*Escherichia coli* and *Ralstonia solanacearum*) as well as Gram-positive bacteria (*Staphylococcus aureus*), with ciprofloxacin and streptomycin used as a positive control, and the fungal target tested was *Fusarium oxysporum*, with ciprofloxacin used as the positive control. Primary screening of all compounds for antimicrobial activity was conducted by the agar filter paper diffusion method. The antimicrobial activities were significantly different when treated with the concentration of 100 μg/mL of compounds **1**–**9** (Figure 5). The experimental strains (*F. oxysporum*, *E. coli*, and *R. solanacearum*) were very sensitive to compounds **2**, **4**, **6**, **7**, and **9**, and their inhibition zones ranged from 15.22 ± 2.11 mm to 23.35 ± 2.45 mm. The inhibition zone diameter of compound **6** was 23.35 ± 2.45 mm against *B. subtilis* (streptomycin 26.16 ± 1.12 mm), and *S.aureus* was not sensitive to all tested compounds. In order to explore the relationship between the concentration and the anti-microbial activity, the minimum inhibitory concentration (MIC) of each compound was calculated by the 96-well plate method.

The minimum Inhibitory concentrations were calculated according to the Clinical and Laboratory Standards Institute (CLSI) method [47]. Compounds **2**, **4**, **7**, and **9** expressed antifungal activities against *F. oxysporum*, with the MIC values ranging from 6.25 to 25.00 μg/mL (Table 2). Compounds **2**, **4**, **5**, **6**, **7**, and **9** showed antibacterial activity, with compounds **4**, **5**, and **9** capable of targeting Gram-negative bacteria (*E. coli* and *R. solanacearum*, MIC = 3.13–12.50 μg/mL). Significantly, the MIC values of compounds **4** and **5** against *R. solanacearum* were equivalent to streptomycin (MIC = 12.50 μg/mL), together with compound **5** against *E. coli* as well as streptomycin (MIC = 3.13 μg/mL). Of note, the MIC value of compound **9** against *R. solanacearum* was 6.25 μg/mL, which showed that the compound had significant potential for further exploitation as an antibacterial compound. None of the *H. lauricola* terpenoids tested expressed any growth inhibitory effects on the Gram-positive *S. aureus* bacterium tested.

### 3.6. Antioxidant Activity

The ability of the *H. lauricola* terpenoids to scavenge free-radical oxidants was assayed using DPPH reagents and a superoxide radical scavenging kit as detailed in the Materials and Methods section. Versus DPPH, the scavenging ability of compounds **2**, **4**, and **6** were roughly equivalent to the positive control (89.98%, BHT) and a little lower than the V_C_ positive control, reaching a final scavenging clearance between 80 and 90% at 100 μg/mL of the test compound (Figure 6A). In contrast, the scavenging ability of these same compounds towards superoxide was generally lower than the control, reaching only 45–66% at 100 μg/mL (Figure 6B). Calculated IC_50_ values confirmed compound **2** as the most potent antioxidant with IC_50_ vs. DPPH = 3.5 ± 1.4 μg/mL and IC_50_ vs. BHT = 25 ± 1.4 μg/mL (Table 3).

## 4. Discussion

*Harringtonia lauricola* displays a unique lifestyle being both a mutualist of *Xyloborus* ambrosia beetles, i.e., acting as their sole food source, but also a potentially devastating plant pathogen to susceptible trees. Little, however, is known concerning the range and nature of secondary metabolites produced by this fungus. Growth of *H. lauricola* on RSM medium produced a particular yeast-like odor and a morphological pattern different from growth on PDB media. These results are consistent with observations of *H. lauricola* colonization of plant hosts resulting in the production of ethanol and other alcohols (putatively via alcohol dehydrogenase activity), which act as attractants for other beetles [48,49,50]. As the fermentation progressed during the 28 d incubation period, oxygen levels were reduced, with anaerobic conditions enhancing fungal alcohol-producing metabolic activity. Within the host tree beetle–fungal galleries, it is speculated that pores seen in the substrate may be overflow channels for metabolically generated carbon dioxide, volatile alkaline nitrogen oxides, and/or other organic compounds [51,52,53]. For some beetle–fungal symbiont pairings (including Xyloborus–*H. lauricola*), galleries can contain multiple fungal members. For Ophiostomaid fungi, fungal volatiles of mutualists with bark beetles have been shown to vary in the presence of other species of mutualists, with similarities potentially reflecting a common ecological niche and differences in species-specific adaptations [54]. VOCs identified included acetoin, ethyl, and phenethyl acetate, and various alcohols, although terpenes were not directly examined. Intriguingly, some of these fungal volatiles can act as carbon sources and/or semiochemicals mediating interspecies interactions as part of the bark beetle fungal symbiont consortium [55]. Ophiostomaid fungi have also been shown to be able to produce host beetle pheromone and/or semiochemical compounds (e.g., the beetle antiaggregation hormone verbenone), particularly in response to host tree chemical compounds [56].

Here, we have identified a series of terpene compounds produced by *H. lauricola*. How these terpenes may affect the chemical ecology of *H. lauricola* and its beetle partner within tree galleries is beyond the scope of this proposal; however, using a series of well-known bacterial and fungal target species, we show significant antimicrobial activity for several of the compounds. With respect to antimicrobial activity, four of the compounds (**2**, **4**, **7**, and **9**) exhibited antifungal activity against the plant pathogenic fungus *F. oxysporum*. Several of the isolated compounds (**2**, **4**–**7** towards *E. coli* and **4**–**6**, **9** towards *R. solanacearum*) showed antibacterial activity against Gram-negative bacteria; however, as can be noted, these sets do not completely overlap, suggesting specific antibacterial targets for some of the compounds. Four *H. lauricola* terpenes (**2**, **6**, **7**, and **9**) showed antibacterial activity towards Gram-positive bacteria (note against *B. subtilis* but not *S. aureus*), indicating that some of these compounds are active against both Gram-positive and Gram-negative bacteria although with target-specific susceptibility. Fungal diterpenes from *Sarcodon scabrosus*, including compounds sarcodonin L, allocyathin B2, sarcodonin G, and sarcodonin L, have also been shown to possess antibacterial activities [57]. Terpenoids with the same skeleton often show different biological activities. Via comparisons between the structures of different labdane diterpenoids, a carbonyl group (C-8), a hydroxyl group (C-19), and a lactone ring have been shown to be the main factors affecting antibacterial activity [58,59], consistent with the structural features and activities of the diterpenoids characterized in this report. In addition, because terpenoids participate in the energy metabolism of mitochondrial intima, they can also indirectly affect the accumulation of energy, including by inhibiting the growth of mycelia and/or producing fungistatic effects [60,61].

In order to provide a broader dataset for potential bioprospecting, we sought to examine any antiproliferative effects of any of the *H. lauricola* compounds isolated. Several labdane diterpenoids have previously been shown to be able to inhibit cell proliferation by inducing apoptosis [62,63]. These effects appear to involve perturbation of mitochondrial membrane potential and increasing intracellular ROS levels. Furthermore, some diterpenoids can cause cell cycle arrest in the G2/M phase at low concentrations and G0/G1 phase arrest at high concentrations [64]. In addition, via enzymatic engineering, the structure of several labdane diterpenoids have been modified to obtain products with enhanced activity [65]. *H. lauricola* diterpene compounds **4**, **5**, **6**, and **7** all showed antiproliferative activity towards lung, breast, and liver cancer cell lines, with compound **9** showing antiproliferative activity towards a liver cancer cell line alone. Terpenoids being structurally altered can lead to their antitumor activities being enhanced or diminished [66]. Our data also show that some of the *H. lauricola* diterpenes show higher activity towards DPPH, which acts as an electron transfer (SET-type), as opposed to superoxide, which is a hydrogen atom transfer (HET-type) free radical [67]. This suggests that the polyhydroxy structure of these terpenoids might have some preferential activity against SET-type radicals. Such scavenging of intracellular reactive oxygen species represents the activity of a direct antioxidant [68,69]. In combination with the evaluation of anti-tumor activity, we found that several of the isolated *H. lauricola*-derived terpenoids display good inhibitory activity on the proliferation of liver cancer cells and also have good antioxidant activity, suggesting a potential relationship between the two activities as the liver is involved in organismal antioxidant process [70,71]. Studies have also shown that in the oxidative damage model of liver cancer cell etiology, antioxidants can indirectly resist oxidative damage of cells through the expression of antioxidant enzymes and genes; that is, via induction of cellular oxidative stress responses [72]. The terpenoids we obtained may not only scavenge cellular free radicals, but could also be acting to enhance endogenous antioxidant defense systems (antioxidant enzymes and glutathione system), and hence their protective mechanisms against oxidatively damaged cells deserve further investigation. The overall characterization of the *H. lauricola* diterpenoids and the various activities examined herein suggest that they may play important roles in inhibiting competing microbes (e.g., within the tree gallery), providing resistance against oxidative stress, and even potentially enhancing the nutritive value of the fungus for its beetle host.

## 5. Conclusions

Here, we show that *H. lauricola*, when cultivated in brown rice and Lauraceae species sawdust, produces abundant bioactive compounds, and a total of six labdane diterpenoids and three hopane triterpenes were isolated from fungal cultures. These compounds (**1**–**9**) were isolated from *H. lauricola*, and even more broadly from Ophiostomatales, for the first time. To determine the potential biological and biopharmaceutical function(s) of these substances, all of the compounds were evaluated for antibacterial, antifungal, antiproliferative, and antioxidant bioactivities. Compounds **2**, **4**, and **6** showed potential antitumor, antibacterial and antioxidant activities. The compounds characterized were diterpenoids, and the various activities characterized herein suggest that they may play important roles in inhibiting competing microbes (e.g., within the tree gallery), providing resistance against oxidative stress, and even potentially enhancing the nutritive value of the fungus for its beetle host. Our study expands the range of biological activities of these terpenoids, providing a reference for the development and utilization of secondary metabolites, and provides the first clues as to the potential contributions of secondary metabolites to the unique lifestyle of this fungus, including the ability to grow as a saprophyte, plant pathogen, and insect (beetle) symbiont. The active compounds described are all small-molecular-weight terpenoids and aromatic ketones. Our analyses of these compounds indicate the significant structural diversity of active metabolites found in insects, plants and fungi, which can be a rich reservoir for biopharmaceutical discovery and applications.

## Figures and Tables

**Figure 1 jof-09-01175-f001:**
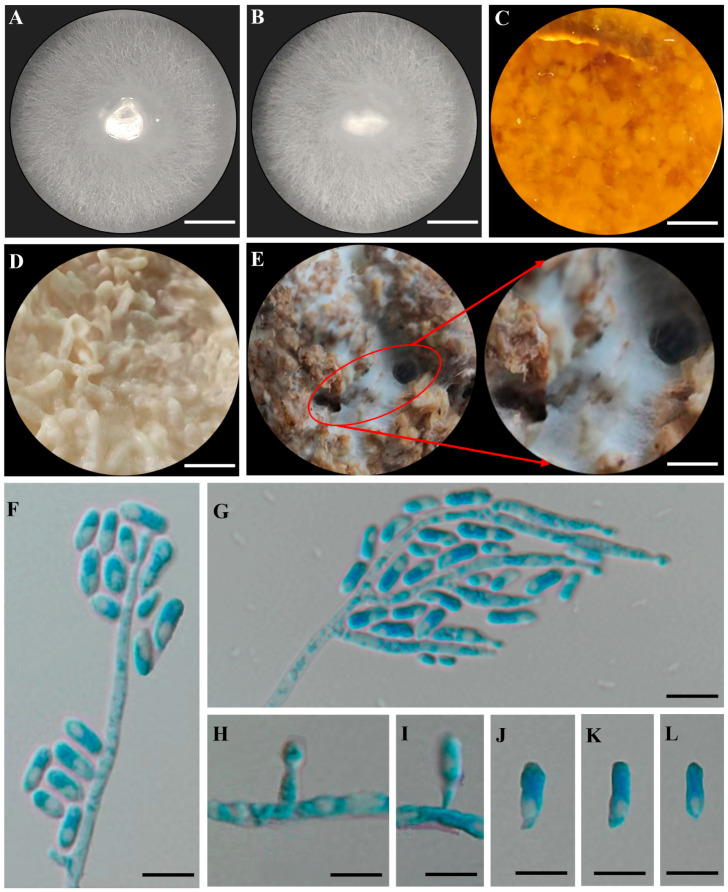
Morphological characteristics of *Harringtonia lauricola* CGMCC3.24979. (**A**,**B**) represent the front and back of CGMCC3.24979 grown on PDA plates. (**C**) CGMCC3.24979 incubated in PDB (26 °C, 15 d). (**D**) CGMCC3.24979 grown on RGM (26 °C, 15 d). (**E**) CGMCC3.24979 grown on RSM (26 °C, 15 d); the red ellipse box indicates the zoom area (zoomed 3× larger than the original image). (**F**,**G**) Conidiophores, conidiogenous cells, and conidia. (**H**,**I**) Conidia formed on conidiogenous cells. (**J**–**L**) Conidia. Scale bars: (**A**,**B**) = 1 cm, (**C**–**E**) = 1.5 cm, (**F**–**I**) = 10 μm, (**J**–**L**) = 5 μm.

**Figure 2 jof-09-01175-f002:**
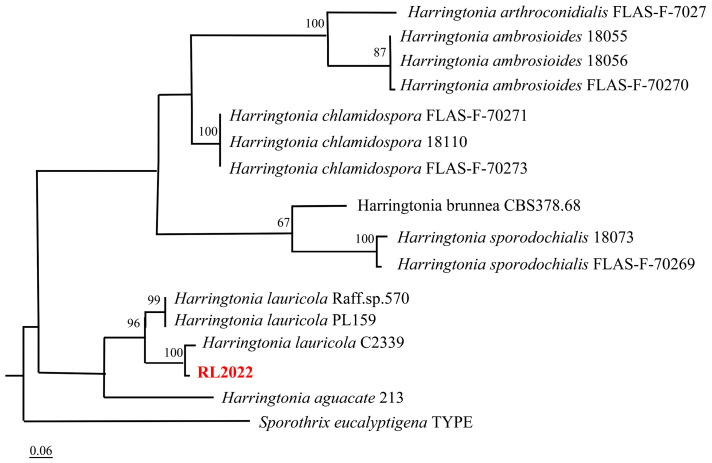
Maximum likelihood tree of *Harringtonia* species constructed using multigene of ITS, LSU, and *β*-Tubulin based on RAxML analysis. Test strain is highlighted in red.

**Figure 3 jof-09-01175-f003:**
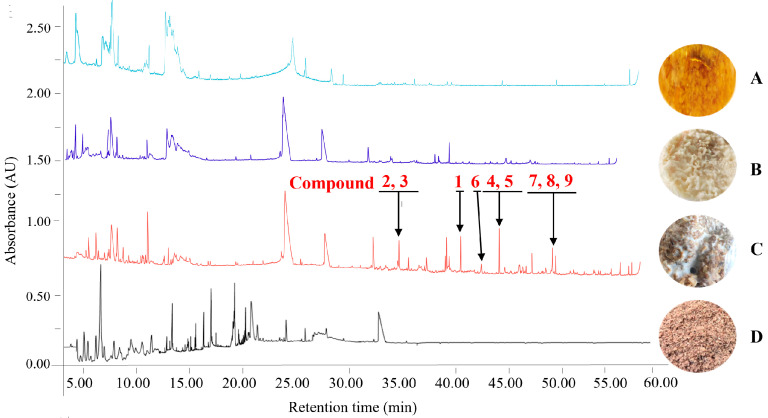
Metabolite profiles of *Harringtonia lauricola* cultivated with small-scale fermentation on various culture media and analyzed by HPLC analysis after extraction. (**A**) PDB, (**B**) RGM, (**C**) RSM, (**D**) Control: RSM (no fungus). The black arrows point to the peaks of isolated compounds **1**–**9** from *H. lauricola* in Section 3.3.

**Figure 4 jof-09-01175-f004:**
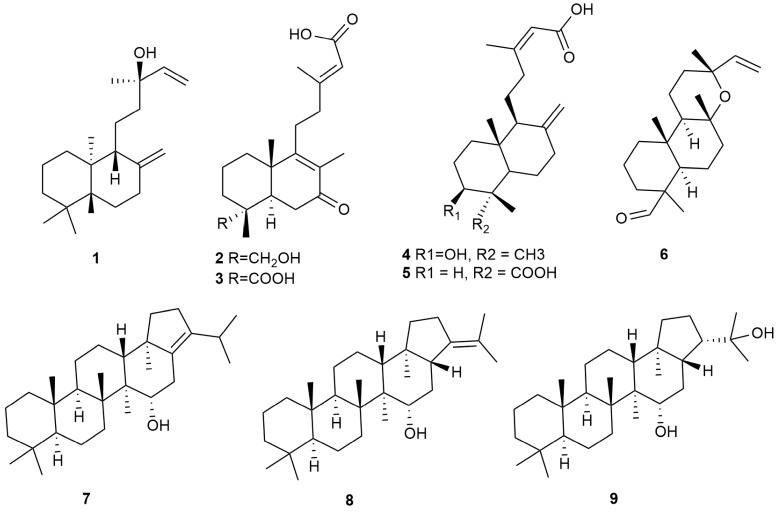
The chemical structures of compounds **1**–**9** isolated from *Harringtonia lauricola*. Compounds **1**–**6** are labdane diterpenoids and compounds **7**–**9** are hopane triterpenoids.

**Figure 5 jof-09-01175-f005:**
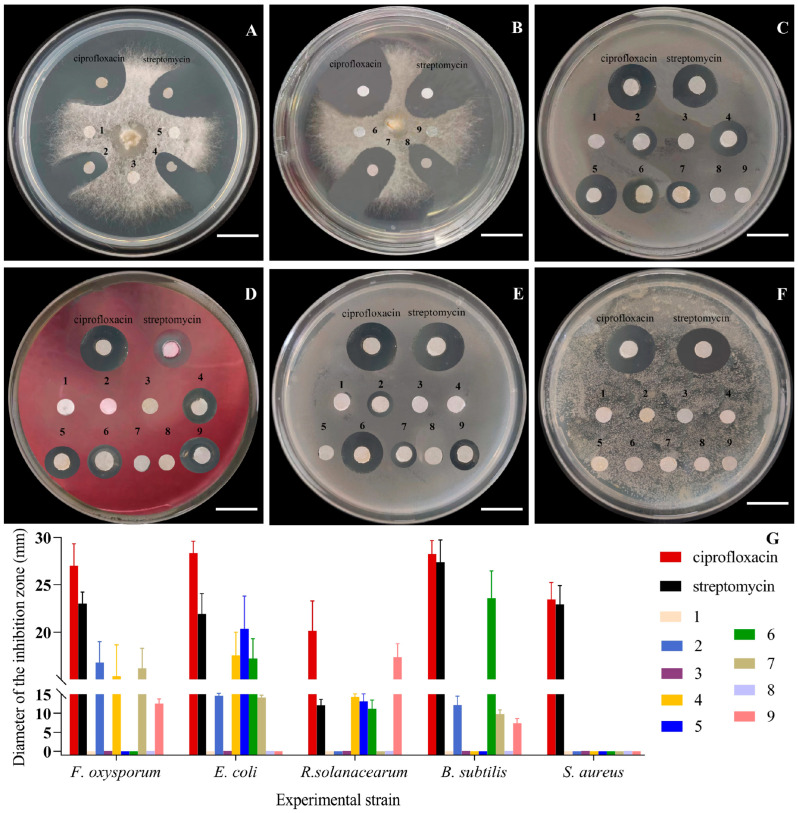
Antimicrobial activities of *H. lauricola* terpenoids (100 μg/mL of the test compounds). (**A**) and (**B**) *F. oxysporum.* (**C**) *E. coli.* (**D**) *R. solanacearum.* (**E**) *B. subtilis.* (**F**) *S. aureus.* (**G**) The inhibition zone diameter of compounds **1**–**9**. Scale bars: (**A**–**F**) = 2 cm. Evaluation standard of agar filter paper diffusion experiment: Extremely sensitive (diameter of inhibition zone > 20 mm). Very sensitive (diameter: 15–20 mm). Moderately sensitive (diameter: 10–15 mm). Sensitive (diameter: 7–10 mm). Not sensitive (diameter < 7 mm).

**Figure 6 jof-09-01175-f006:**
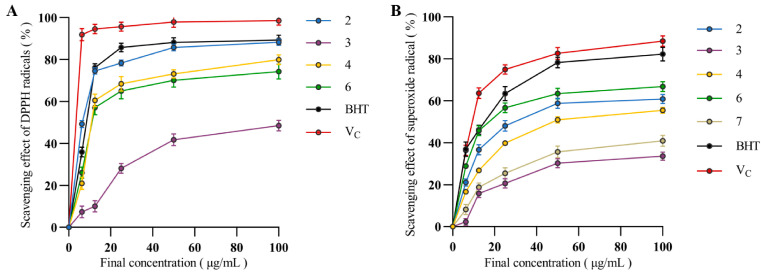
Antioxidant activity of compounds **1**–**9**. (**A**) DPPH free radical scavenging activity; compounds **1**, **5**, **7**, **8**, and **9** have no antioxidant activity. (**B**) Superoxide radical scavenging activity; compounds **1**, **5**, **8**, and **9** have no antioxidant activity. BHT and V_C_ were used as positive controls. Data are expressed as the mean ± SD, (*n* = 3).

**Table 1 jof-09-01175-t001:** Antiproliferative activity of *H. lauricola* terpenoids.

Compounds	In Vitro Antiproliferative IC_50_ (μM)		
	MCF-7 (Breast)	HepG2 (Liver)	A549 (Lung)
**1**	-	-	-
**2**	23.16 ± 0.96	16.90 ± 0.68	19.88 ± 1.02
**3**	-	-	-
**4**	26.06 ± 0.63	15.42 ± 0.52	18.43 ± 0.74
**5**	21.48 ± 0.31	12.54 ± 0.33	15.11 ± 0.41
**6**	27.31 ± 0.59	38.46 ± 0.38	27.53 ± 0.45
**7**	48.61 ± 0.22	53.65 ± 0.34	36.52 ± 0.12
**8**	-	-	-
**9**	-	18.81 ± 1.25	-
cisplatin	3.10 ± 0.57	2.40 ± 0.42	1.50 ± 0.81

IC_50_: half-maximal inhibitory concentration; “-”: IC_50_ > 100 (no anti-tumor effect). Cisplatin was used as a positive control. Data are expressed as mean ± SD from three experiments (*p* < 0.05).

**Table 2 jof-09-01175-t002:** Antimicrobial activities of *H. lauricola* terpenoids.

Compounds	Minimal Inhibitory Concentration (MIC, μg/mL)				
	*F. oxysporum*	*E. coli*	*R. solanacearum*	*B. subtilis*	*S. aureus*
**1**	-	-	-	-	-
**2**	6.25	12.50	-	50.00	-
**3**	-	-	-	-	-
**4**	12.50	6.25	12.50	-	-
**5**	-	3.13	12.50	-	-
**6**	-	6.25	25.00	3.13	-
**7**	6.25	12.5	-	50.00	-
**8**	-	-	-	-	-
**9**	25.00	-	6.25	25.00	-
ciprofloxacin	0.78	0.78	3.13	0.78	1.56
streptomycin	1.56	3.13	12.50	0.78	1.56

“-”: no antimicrobial activity. Ciprofloxacin and streptomycin were used as positive controls.

**Table 3 jof-09-01175-t003:** Antioxidant activity (IC_50_ values) of *H. lauricola* terpenoids tested against DPPH and superoxide.

Compounds	IC_50_ (μg/mL)	
	DPPH Radical Scavenging	Superoxide Radical Scavenging
**1**	-	-
**2**	3.45 ± 1.41	25.38 ± 1.44
**3**	>50.00	>50.00
**4**	14.04 ± 1.42	43.31 ± 1.46
**5**	-	-
**6**	12.37 ± 1.27	22.87 ± 1.35
**7**	-	>50.00
**8**	-	-
**9**	-	-
BHT	6.42 ± 1.93	14.43 ± 0.38
V_C_	1.70 ± 1.11	8.518 ± 0.57

IC_50_: half inhibitory concentration, (*p* < 0.05). “-”: no antioxidant activity.

## Data Availability

Data are contained within the article.

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
