# Peer review of "Characterization of Terpenoids from the Ambrosia Beetle Symbiont and Laurel Wilt Pathogen Harringtonia lauricola"

_jof, 2023, doi:10.3390/jof9121175_

Round 1
Reviewer 1 Report
Comments and Suggestions for Authors
Comments to the manuscript “Characterization of terpenoids from the ambrosia beetle symbiont and laurel wilt pathogen, Harringtonia lauricola” by Zhu et al.
General comment
The submitted manuscript analyses the terpenoids production by a fungal isolate identified as Harringtonia lauricola. The strain identification was conducted by morphological and three gene phylogenetic analysis. The secondary metabolites production was analyzed in three different growth media. The best medium growth for metabolite production of the studied strain was rice mixed with Cinnamomum camphora sawdust (RSM), which was used for large scale fermentation assays to isolate and identify the metabolites produced. Nine different terpenoids (6 labdane diterpenoids (1-6) and 3 hopane triterpenes (7-9)) were isolated and identified. The antiproliferative, antimicrobial, and antioxidant activities of the identified compounds were tested.
In general, the submitted manuscript is well written, the methodology is accurately described, and the results are clearly explained.
Thus, I consider that the document is suitable for its publication in the Journal of Fungi after a minor review. Below are some specific comments for the authors' consideration.
Specific comments:
1. It should be relevant to include information regarding site and date of the fungal strain analyzed.
2. Please, describe the best evolutive model(s) used to reconstruct the phylogenetic tree with the three genetic regions used.
3. How was it possible to include in the phylogenetic analysis (Figure 2) to strains/species that did not have the ITS region or the LSU gene (Supplementary Table S2)?
4. I think the section that needs a major restructure is the Discussion. The authors limit the discussion to the establishment of general hypotheses about the structures and mechanisms of the different biological activities of terpenoids. But there are several aspects that would be important to discuss such as:
i. Why did the RSM medium induce a greater production of metabolites than the other two mediums evaluated? Are there other media based on plant residues in which the production of secondary metabolites, particularly terpenes, has been described?
ii. In which fungal genera/species were previously reported the terpenes found in H. lauricola? Are these fungi closely related to in H. lauricola or phylogenetically distinct? (the answer can indicate how restricted, or not, are the synthesis of such compounds un the Fungal kindgdom).
iii. Terpene production has been described in other ophiostomatoid fungal species (Cale et al. 2019. Fungal Ecology. 39, 285-295). Can the terpenes described in H. lauricola be associated with the functions described for other terpenes of the ophiostomatoid fungi?
iv. How do IC50 (antiproliferative activity) MIC (antimicrobial activity) values compare to other terpenes produced by fungi?
I consider that the discussion should address comparisons with previous literature such as those exemplified above.
Author Response
Dear Editors and Reviewers:
Thank you for your letter and comments relating to our manuscript entitled “Characterization of terpenoids from the ambrosia beetle symbiont and laurel wilt pathogen, Harringtonia lauricola” (ID: jof-2720139). The comments were very helpful in revising and improving our manuscript as well as emphasizing the significance to our research. We have read the comments carefully and made corrections accordingly. Revised portions are marked in blue in the manuscript. The main corrections in the paper and our responses to the reviewer’s comments are given below. We hope that the revisions in the manuscript and our accompanying responses will be sufficient to make our manuscript suitable for publication in the Journal of Fungi.
Responses to the comments of the reviewers:
Reviewer 1#
Comments 1: It should be relevant to include information regarding site and date of the fungal strain analyzed.
Response 1: We have added information regarding site and date of the fungal strain analyzed.
The fungal strain RL2022 was initially isolated from ambrosia beetles collected in Fuzhou National Forest Park in Fujian Province, China on 3 July 2022, deposited in the China General Microbiological Culture Collection Center (CGMCC), and originally identified as Harringtonia sp.
Comments 2: Please, describe the best evolutive model(s) used to reconstruct the phylogenetic tree with the three genetic regions used.
Response 2: We have described GTRGAMMA model used to reconstruct the phylogenetic tree.
Phylogenetic analyses were performed using maximum likelihood (ML) in RAxML (8.2.10, GTRGAMMA model, bootstrap=1000) [30]
- Stamatakis, A. RAxML version 8: a tool for phylogenetic analysis and post-analysis of large phylogenies. Bioinformatics. 2014, 30, 1312-1313. doi: 10.1093/bioinformatics/btu033
Comments 3: How was it possible to include in the phylogenetic analysis (Figure 2) to strains/species that did not have the ITS region or the LSU gene (Supplementary Table S2)?
Response 3: We used their previous placement made by others based on morphological characteristics, however, for sake of clarity these have been removed from the tree and the tree has been redrawn.
Comments 4: I think the section that needs a major restructure is the Discussion. The authors limit the discussion to the establishment of general hypotheses about the structures and mechanisms of the different biological activities of terpenoids.
Response: Aspects of both the Introduction and Discussion have undergone major revisions.
But there are several aspects that would be important to discuss such as:
Comments 4i: Why did the RSM medium induce a greater production of metabolites than the other two mediums evaluated? Are there other media based on plant residues in which the production of secondary metabolites, particularly terpenes, has been described?
Response 4i: The RSM medium with additional active precursors can promote the synthesis and secretion of secondary substances by simulating the (microbial) habitat/habitat environment, resulting in the production of new secondary metabolites because possible silent biosynthetic gene clusters of microorganisms can be activated. As described, other media based on plant residues, when compared with pure cultures and co-cultures, could not only increase the diversity of secondary metabolites but also improve their biological activity and yield (Li et al., 2023).
Li, H.T.; Zhou, H.; Ding, Z.T. Advances in producing novel active secondary metabolites by co-culture of microorganisms. Journal of Yunnan University.2023, 45, 493-512.doi: 10.7540/j.ynu.20220418
Comments 4ii: In which fungal genera/species were previously reported the terpenes found in H. lauricola? Are these fungi closely related to in H. lauricola or phylogenetically distinct? (the answer can indicate how restricted, or not, are the synthesis of such compounds un the Fungal kindgdom).
Response 4ii: Simon et al. (2017) have reported that terpenes are produced in Harringtonia lauricola (Table A).
Simon, A.G.; Mills, D.K.; Furton, K.G. Chemotyping the temporal volatile organic compounds of an invasive fungus to the United States, Raffaelealauricola. J.Chromatogr. A. 2017, 1487, 72-76. doi: 10.1016/j.chroma.2017.01.065
Table A. Terpenes produced in Harringtonia lauricola
Compounds |
Organism species |
1 |
Chiloscyphus polyanthus |
2 |
Forsythia suspensa |
3 |
Forsythia suspensa |
4 |
Fruits of Forsythia suspensa |
5 |
Araucaria cunninghamii |
6 |
Abies nukiangensis |
7 |
Drynariafortunei |
8 |
Lichens and fungi |
9 |
Entomopathogenic fungus Hypocrella sp. |
Comments 4iii: Terpene production has been described in other ophiostomatoid fungal species (Cale et al. 2019. Fungal Ecology. 39, 285-295). Can the terpenes described in H. lauricola be associated with the functions described for other terpenes of the ophiostomatoid fungi? I consider that the discussion should address comparisons with previous literature such as those exemplified above.
Response 4iii: We have revised the Discussion to consider the point raised by the reviewer.
Comments 4iv: How do IC50 (antiproliferative activity) MIC (antimicrobial activity) values compare to other terpenes produced by fungi?
Response 4iv: The Discussion has been edited to consider the reviewer’s point.
Harringtonia lauricola displays a unique lifestyle being both a mutualist of Xyloborus ambrosia beetles, i.e., acting as their sole food source, but also a potentially devastating plant pathogen to susceptible trees. Little, however, is known concerning the range and nature of secondary metabolites produced by this fungus. Growth of H. lauricola on RSM medium produced a particular yeast-like odor and a morphological pattern different from growth on PDB media. These results are consistent with observations of H.lauricola colonization of plant hosts results in the production of ethanol and other alcohols (putatively via alcohol dehydrogenase activity), which act as attractants for other beetles [48-50]. As the fermentation progressed during the 28 d incubation period, oxygen levels were reduced, with anaerobic conditions enhancing fungal alcohol-producing metabolic activity. Within the host tree beetle-fungal galleries, it is speculated that pores seen in the substrate may be overflow channels for metabolically generated carbon dioxide, volatile alkaline nitrogen oxides, and/or other organic compounds [51-53]. For some beetle-fungal symbiont pairings (including Xyloborus-H. lauricola), galleries can contain multiple fungal members. For Ophiostomaid fungi, fungal volatiles of mutualists with bark beetles have been shown to vary in the presence of other species of mutualists, with similarities potentially relecting a common ecological niche and differences species-specific adaptations [54]. VOCs identified included acetoin, ethyl- and phenethyl acetate, and various alcohols, although terpenes were not diretly examined. Intriguingly, some of these fungal volatiles can act as carbon sources and/or semiochemicals mediating interspecies interactions as part of the bark beetle fungal symbiont consortium [55]. Ophiostomaid fungi have also been shown to be able to produce host beetle pheromone and/or semiochemical compounds (e.g., the beetle antiagreegation hormone, verbenone), particularly in response to host tree chemical compounds [56].
Here, we have identified a series of terpene compounds produced by H. lauricola. How these terpenes may affect the chemical ecology of H. lauricola and its beetel partner within tree galleries, is beyond the scope of this proposal, however, using a sereis of well known bacterial and fungal target species we show signficant antimicrobial activity for several of the compounds. With respect to antimicrobial activity, four of the compounds (2, 4, 7, and 9) exhibited anti-fungal activity against the plant pathogenic fungus, F. oxysporum. Several of the isolated compounds (2, 4-7 towards E. coli and 4-6, 9 towards R. solanacearum) showed antibacterial activity against gram-negative bacteria, however, as can be noted, these sets do not completely overlap suggesting specific antibacterial targets for some of the compounds. Four H. lauricola terpenes (2, 6, 7, and 9) showed antibacterial activity towards gram-positive bacteria (note against B. subtilis but not S. aureus), indicating that some of these compounds are active against both gram-postivie and gram-netative bacteria although with target specific susceptibilty. Fungal diterpened from Sarcodon scabrosus, including compounds sarcodonin L, allocyathin B2, sarcodonin G and sarcodonin L have also been shown to possess antibacterial activities [57]. Terpenoids with the same skeleton often show different biological activities. Via comparisons between the structures of different labdane diterpenoids, a carbonyl group (C-8), a hydroxyl group (C-19) and a lactone ring have been shown to be the main factors affecting antibacterial activity [58, 59], consistent with the structural features and activities of the diterpenoids characterized in this report. In addition, because terpenoids participate in the energy metabolism of mitochondrial intima, they can also indirectly affect the accumulation of energy, including by inhibiting the growth of mycelia and/or producing fungistatic effects [60, 61].
In order to provide a broader dataset for potential bioprospecting, we sought to examine any antiproliferative effects of any of the H. lauricola compounds isolated. Several labdane diterpenoids have previously been shown to be able to inhibit cell proliferation by inducing apoptosis [62, 63]. These effects appear to involve perturbation of mitochondrial membrane potential and increasing intracellular ROS levels. Furthermore, some diterpeniods can cause cell cycle arrest in the G2/M phase at low concentrations and G0/G1 phase arrest at high concentrations [64]. In addition, via enzymatic engineering, the structure of several labdane diterpenoids have been modified to obtain products with enhanced activity [65]. H. lauricola diterpene compounds 4, 5, 6, and 7, all showed antiproliferative activity towards lung, breast, and liver cancer cell lines, with compound 9 showing antiproliferative activity towards a liver cancer cell line, alone. Terpenoids structurally altered can lead to their antitumor activities getting enhanced or diminished [66]. Our data also show that some of the H. lauricola diterpenens show higher activity towards DPPH, which acts as an electron transfer (SET-type), as opposed to superoxide, which is a hydrogen atom transfer (HET-type) free radical [67]. This suggests that the polyhydroxy structure of these terpenoids might have some preferential activity against SET-type radicals. Such scavenging of intracellular reactive oxygen species represents the activity of a direct antioxidant [68, 69]. In combination with the evaluation of anti-tumor activity, we found that several of the isolated H. lauricola derived terpenoids display good inhibitory activity on the proliferation of liver cancer cells also have good antioxidant activity, suggesting a potential relationship between the two activities as the liver is involved in organismal antioxidant process [70, 71]. Studies have also shown that in the oxidative damage model of liver cancer cell etiology, antioxidants can indirectly resist oxidative damage of cells through the expression of antioxidant enzymes and genes, that is, via induction of cellular oxidative stress responses [72]. The terpenoids we obtained may not only scavengers cellular free radicals, but could also be acting to enhance endogenous antioxidant defense systems (antioxidant enzymes and glutathione system), and hence their protective mechanisms against oxidatively damaged cells deserve further investigation. The overall characterization of the H. lauricola diterpenoids and the various activties examined herein suggest that they may play important roles in inhibiting competing microbes (e.g., within the tree gallery), providing resistance against oxidative stress, and even potenially enhancing the nutritive value of the fungus for its beetle host.
We tried our best to improve the manuscript and made some changes marked in blue in revised paper which will not influence the content and framework of the paper. We appreciate for Editors/Reviewers’ warm work earnestly and hope the revision will meet with your approval. Once again, thank you very much for your comments and suggestions.
Kind regards,
Junzhi Qiu
E-mail address: [email protected]
Reviewer 2 Report
Comments and Suggestions for Authors
The manuscript presents the results of research conducted on compounds isolated from the fungus Harringtonia lauricola. Nine known terpenoid compounds were subjected to various bioassays. The selected compounds showed antiproliferative, antimicrobial, antifungal, antioxidant activities. The manuscript is interesting, but needs some revisions.
Minor remarks
In the abstract there was an error in the description of the ability of the tested compounds to scavenge free radicals. It was first stated that this was related to compounds 4, 5 and 9, and then to compounds 2, 4 and 6. This should be corrected.
In the introduction there is information about the harmfulness of H. lauricola, the search for a way to control it, followed by a statement about the lack of information about the secondary metabolites produced by this fungus. What is the connection? In the text, the authors focused mainly on characterizing the diterpenes isolated from H. lauricola and highlighting their valuable properties. Therefore, the introduction should not be mainly about the harmfulness of the tested fungus, as this suggests a different research trend.
Fig. 5 - it would be advisable to enlarge the graphs. As it stands, they are not very legible.
Item 3.6 - wrongly defined unit in the statement "80%-90% at 100g/ml" and "45%-66% at 100g/ml".
Author Response
Dear Editors and Reviewers:
Thank you for your letter and comments relating to our manuscript entitled “Characterization of terpenoids from the ambrosia beetle symbiont and laurel wilt pathogen, Harringtonia lauricola” (ID: jof-2720139). The comments were very helpful in revising and improving our manuscript as well as emphasizing the significance to our research. We have read the comments carefully and made corrections accordingly. Revised portions are marked in blue in the manuscript. The main corrections in the paper and our responses to the reviewer’s comments are given below. We hope that the revisions in the manuscript and our accompanying responses will be sufficient to make our manuscript suitable for publication in the Journal of Fungi.
Responses to the comments of the reviewers:
Reviewer 2#
Comments 1: In the abstract there was an error in the description of the ability of the tested compounds to scavenge free radicals. It was first stated that this was related to compounds 4, 5 and 9, and then to compounds 2, 4 and 6. This should be corrected.
Response 1: We regret the error and the text has been corrected.
Little is known concerning terpenoids produced by members of the fungal order Ophiostomales with member Harringtonia lauricola having the unique lifestyle of being a beetle symbiont but potentially devastating tree pathogen. Nine known terpenoids, including six labdane diterpenoids (1-6) and three hopane triterpenes (7-9), were isolated from H. lauricola ethyl-acetate (EtOAc) extracts for the first time. All compounds were tested for various in vitro bioactivities. The six compounds, 2, 4, 5, 6, 7 and 9, are described functionally. Compounds 2, 4, 5 and 9 expressed potent antiproliferative activity against the MCF-7, HepG2 and A549 cancer cell lines, with half-maximal inhibitory concentrations (IC50s) ~12.54-26.06 μM. Antimicrobial activity bioassays revealed that compounds 4, 5 and 9 exhibited substantial effects against gram-negative bacteria (Escherichia coli and Ralstonia solanacearum) with minimum inhibitory concentration (MIC) values between 3.13-12.50 μg/mL. Little activity was seen towards gram-positive bacteria for any of the compounds, whereas compounds 2, 4, 7 and 9 expressed antifungal activities (Fusarium oxysporum) with MIC values ranging from 6.25-25.00 μg/mL. Compounds 4, 5 and 9 also displayed free radical scavenging abilities towards 2,2-diphenyl-1-picrylhydrazyl (DPPH) and superoxide (O2-), with IC50 values of compounds 2, 4 and 6 ~3.45-14.04 μg/mL and 22.87-53.31 μg/mL towards DPPH and O2-, respectively. These data provide insight into the biopharmaceutical potential of terpenoids from this group of fungal insect symbionts and plant pathogens.
Comments 2: In the introduction there is information about the harmfulness of H. lauricola, the search for a way to control it, followed by a statement about the lack of information about the secondary metabolites produced by this fungus. What is the connection? In the text, the authors focused mainly on characterizing the diterpenes isolated from H. lauricola and highlighting their valuable properties. Therefore, the introduction should not be mainly about the harmfulness of the tested fungus, as this suggests a different research trend.
Response 2: We have revised the Introduction.
Fungi are well known to be rich in compounds termed secondary metabolites that display an astonishingly diverse array of biological and biopharmaceutical properties [1, 2]. These include varied classes of compounds ranging from those with potential human health relevance, e.g., antimicrobial, anti-cancerous, and immune-modulatory compounds to those exploitable in industries ranging from food and agriculture, bioremediation, and even cosmetics [3-5]. However, it is estimated that there are more than three million fungi in nature, of which humans have discovered less than 8% [6, 7], indicating a rich unexplored diversity of organisms and their bioactive compounds waiting discovery. Terpenoids (isoprenoids derived from five carbon isoprene units) represent a heterogeneous naturally occurring class of compounds most widely studied in plants, where some have been shown to act as phytohormones, antioxidants, in defense and/or community interactions (e.g., attraction of (beneficial) organisms) [8, 9]. In addition, bioprospecting of terpenoids for a wide range of application is a very active field, with significant efforts expended in applications and platforms for terpenoid production [10-13]. Terpenoids have been characterized from numerous fungal sources including a range of Ascomycetes and Basidiomycetes, especially endophytic fungi [14-17]. However, little is known concerning terpenoids from the Ophiostomales fungal order that includes unique plant pathogens and insect symbionts, particualrly with respect to bioactivties with potential biopharmaceutical applications.
Harringtonia lauricola (formerly Raffaelea lauricola, Ascomycota, Ophiostomales, Ophiostomataceae) is an invasive (to the Southeastern United States) beetle borne/vectored plant pathogenic fungus affecting members of the Lauraceae family, responsible for laurel wilt disease[18]. The original vector for the fungus was the (invasive to the US) red bay ambrosia beetle, Xyleborus glabratus, with the insect and its fungal symbiont (H. lauricola) likely introduced from unprocessed wood during transport from Asia to the Eastern United States around the turn of the 21st century [19, 20]. The fungus is stored in specialized beetle structures termed mycangia [21], and when released by the beetle into trees during gallery excavation, the fungus can attack mature and otherwise healthy hosts. H. lauricola has led to the death of over 300 million trees inthe Southeastern United States since its introduction, being responsible for eliminating >85% of redbay trees in endemic regions[22](!!! INVALID CITATION !!! ). Most fungal symbiotic partners of these beetles, however, do not cause significant damage to trees, suggesting unique aspects of H. lauricola and its interaction with the tree host that results in disease.
Significant aspects of the biology of H. lauricola remain to be characterized. Recent examinations of various physiological growth parameters of H. lauricola revealed the potential for cold adaptation (optimal growth temp 15-260C) and pH sensitivity (reduced growth at pH > 8.0), as well as sensitivity to a range of fungicides including various conazoles, prochloraz, dithiocarbamates, andzinc-based fungicides [23]. In addition, growth substrate profiling revealed broad utilization of sulfur and phosphate containing compounds, comparatively restricted carbon-substrate utilization, and rescue of pH and osmotic sensitivities by specific compounds (e.g., amino acids) [24]. Chemotyping of volatile organic compounds from H. lauricola, identified VOC dynamics potentially linked to response of host trees [25]. These included a suite of alchohols, pentaness, hexenes, and heptanes, as well as monoterpene and terpenes, however, no exact determiantion of the latter compounds were made. Thus, overall, there remains limited information on range and/or types of terpenoid secondary metabolites derived from H. lauricola, although genomic analyses indicate a large repertoire of putative secondary metabolite biosyntheticgene clusters [26-28]. Our objectives were to: (1) identify suitable growth and purification conditions for the isolation of terpenes from H. lauricola, and (2) examine the biopharmaceutical potential of isolated compounds with respect to antibacterial, antifungal, antooxidant, and antproliferative activties. We show that when grown in a medium of brown rice coupled with Lauraceae species sawdust, a suite of H. lauricola derived terpenoids could be isolated including labdane diterpenoids andhopane triterpenes. Using a series of cancer cell lines, target bacterial and fungal species, and reaction oxygen species (ROS), the antiproliferative, antibacterial and antioxidant activities of these compounds were subsequently tested in vitro. These data provide a new window into the secondary metabolite repertoire of H. lauricola that might play functional roles in the unique adaptations of this fungus that includes plant pathogenicity and beetle mutualism as well as provide lead compouds for bioprospecting.
Comments 3: Fig. 5 - it would be advisable to enlarge the graphs. As it stands, they are not very legible.
Response 3: We have enlarged Fig.5.
Comments 4: Item 3.6 - wrongly defined unit in the statement "80%-90% at 100 g/ml" and "45%-66% at 100 g/ml”
Response 4: We have changed them to 80%-90% at 100 μg/mL and 45%-66% at 100 μg/mL, respectively.
We tried our best to improve the manuscript and made some changes marked in blue in revised paper which will not influence the content and framework of the paper. We appreciate for Editors/Reviewers’ warm work earnestly and hope the revision will meet with your approval. Once again, thank you very much for your comments and suggestions.
Kind regards,
Junzhi Qiu
E-mail address: [email protected]
Reviewer 3 Report
Comments and Suggestions for Authors
The manuscript of Zhu et al. entitled “Characterization of terpenoids from the ambrosia beetle symbiont and laurel wilt pathogen, Harringtonia lauricola” is well-written and standard organized paper. It concerns some secondary metabolites revealed in the fungus associated with both insects and trees. However, as I think there are some points to improve the manuscript.
The authors focused on terpenoids of Harringtonia lauricola. Their importance was not stressed both in Introduction and Discussion sections.
No aim, objectives of the study were pronounced.
In the Methods the fermentation period was not indicated.
Check medium for large scale fermentation: PDB or RSM.
Why the mycelium not culture liquid was used for extraction?
What was a principle of the purification of metabolites?
How authors do explain the choice the organisms for bioassays? The selected ones seem cannot explain chemical ecology of the fungus.
The title of Figure 1 should be changed. Please, include the fungal name.
What a characteristic smell of the culture? Please, in more detail.
The quality of HPLC chromatograms seems low. The extraction solvent and HPLC conditions were not optimized for the terpenoids.
The mass spectra of compounds 7-9 are of low quality. Please, explain why.
What organisms are known to produce 1-9? What about of comparison with metabolites known for relative fungi for Harringtonia lauricola.
What means “---" in the table 1?
Change Conclusions accordingly to the aims of the manuscript.
Comments on the Quality of English LanguageEnglish more or less fine. Some mistakes can be found.
Author Response
Dear Editors and Reviewers:
Thank you for your letter and comments relating to our manuscript entitled “Characterization of terpenoids from the ambrosia beetle symbiont and laurel wilt pathogen, Harringtonia lauricola” (ID: jof-2720139). The comments were very helpful in revising and improving our manuscript as well as emphasizing the significance to our research. We have read the comments carefully and made corrections accordingly. Revised portions are marked in blue in the manuscript. The main corrections in the paper and our responses to the reviewer’s comments are given below. We hope that the revisions in the manuscript and our accompanying responses will be sufficient to make our manuscript suitable for publication in the Journal of Fungi.
Responses to the comments of the reviewers:
Reviewer 3#
Comments 1: The authors focused on terpenoids of Harringtonia lauricola. Their importance was not stressed both in Introduction and Discussion sections.
Response 1: The text has been edited to highlight their importance.
Introduction:
Fungi are well known to be rich in compounds termed secondary metabolites that display an astonishingly diverse array of biological and biopharmaceutical properties [1, 2]. These include varied classes of compounds ranging from those with potential human health relevance, e.g., antimicrobial, anti-cancerous, and immune-modulatory compounds to those exploitable in industries ranging from food and agriculture, bioremediation, and even cosmetics [3-5]. However, it is estimated that there are more than three million fungi in nature, of which humans have discovered less than 8% [6, 7], indicating a rich unexplored diversity of organisms and their bioactive compounds waiting discovery. Terpenoids (isoprenoids derived from five carbon isoprene units) represent a heterogeneous naturally occurring class of compounds most widely studied in plants, where some have been shown to act as phytohormones, antioxidants, in defense and/or community interactions (e.g., attraction of (beneficial) organisms) [8, 9]. In addition, bioprospecting of terpenoids for a wide range of application is a very active field, with significant efforts expended in applications and platforms for terpenoid production [10-13]. Terpenoids have been characterized from numerous fungal sources including a range of Ascomycetes and Basidiomycetes, especially endophytic fungi [14-17]. However, little is known concerning terpenoids from the Ophiostomales fungal order that includes unique plant pathogens and insect symbionts, particualrly with respect to bioactivties with potential biopharmaceutical applications.
Harringtonia lauricola (formerly Raffaelea lauricola, Ascomycota, Ophiostomales, Ophiostomataceae) is an invasive (to the Southeastern United States) beetle borne/vectored plant pathogenic fungus affecting members of the Lauraceae family, responsible for laurel wilt disease[18]. The original vector for the fungus was the (invasive to the US) red bay ambrosia beetle, Xyleborus glabratus, with the insect and its fungal symbiont (H. lauricola) likely introduced from unprocessed wood during transport from Asia to the Eastern United States around the turn of the 21st century [19, 20]. The fungus is stored in specialized beetle structures termed mycangia [21], and when released by the beetle into trees during gallery excavation, the fungus can attack mature and otherwise healthy hosts. H. lauricola has led to the death of over 300 million trees inthe Southeastern United States since its introduction, being responsible for eliminating >85% of redbay trees in endemic regions[22](!!! INVALID CITATION !!! ). Most fungal symbiotic partners of these beetles, however, do not cause significant damage to trees, suggesting unique aspects of H. lauricola and its interaction with the tree host that results in disease.
Significant aspects of the biology of H. lauricola remain to be characterized. Recent examinations of various physiological growth parameters of H. lauricola revealed the potential for cold adaptation (optimal growth temp 15-260C) and pH sensitivity (reduced growth at pH > 8.0), as well as sensitivity to a range of fungicides including various conazoles, prochloraz, dithiocarbamates, andzinc-based fungicides [23]. In addition, growth substrate profiling revealed broad utilization of sulfur and phosphate containing compounds, comparatively restricted carbon-substrate utilization, and rescue of pH and osmotic sensitivities by specific compounds (e.g., amino acids) [24]. Chemotyping of volatile organic compounds from H. lauricola, identified VOC dynamics potentially linked to response of host trees [25]. These included a suite of alchohols, pentaness, hexenes, and heptanes, as well as monoterpene and terpenes, however, no exact determiantion of the latter compounds were made. Thus, overall, there remains limited information on range and/or types of terpenoid secondary metabolites derived from H. lauricola, although genomic analyses indicate a large repertoire of putative secondary metabolite biosyntheticgene clusters [26-28]. Our objectives were to: (1) identify suitable growth and purification conditions for the isolation of terpenes from H. lauricola, and (2) examine the biopharmaceutical potential of isolated compounds with respect to antibacterial, antifungal, antooxidant, and antproliferative activties. We show that when grown in a medium of brown rice coupled with Lauraceae species sawdust, a suite of H. lauricola derived terpenoids could be isolated including labdane diterpenoids andhopane triterpenes. Using a series of cancer cell lines, target bacterial and fungal species, and reaction oxygen species (ROS), the antiproliferative, antibacterial and antioxidant activities of these compounds were subsequently tested in vitro. These data provide a new window into the secondary metabolite repertoire of H. lauricola that might play functional roles in the unique adaptations of this fungus that includes plant pathogenicity and beetle mutualism as well as provide lead compouds for bioprospecting.
Discussion:
Harringtonia lauricola displays a unique lifestyle being both a mutualist of Xyloborus ambrosia beetles, i.e., acting as their sole food source, but also a potentially devastating plant pathogen to susceptible trees. Little, however, is known concerning the range and nature of secondary metabolites produced by this fungus. Growth of H. lauricola on RSM medium produced a particular yeast-like odor and a morphological pattern different from growth on PDB media. These results are consistent with observations of H.lauricola colonization of plant hosts results in the production of ethanol and other alcohols (putatively via alcohol dehydrogenase activity), which act as attractants for other beetles [48-50]. As the fermentation progressed during the 28 d incubation period, oxygen levels were reduced, with anaerobic conditions enhancing fungal alcohol-producing metabolic activity. Within the host tree beetle-fungal galleries, it is speculated that pores seen in the substrate may be overflow channels for metabolically generated carbon dioxide, volatile alkaline nitrogen oxides, and/or other organic compounds [51-53]. For some beetle-fungal symbiont pairings (including Xyloborus-H. lauricola), galleries can contain multiple fungal members. For Ophiostomaid fungi, fungal volatiles of mutualists with bark beetles have been shown to vary in the presence of other species of mutualists, with similarities potentially relecting a common ecological niche and differences species-specific adaptations [54]. VOCs identified included acetoin, ethyl- and phenethyl acetate, and various alcohols, although terpenes were not diretly examined. Intriguingly, some of these fungal volatiles can act as carbon sources and/or semiochemicals mediating interspecies interactions as part of the bark beetle fungal symbiont consortium [55]. Ophiostomaid fungi have also been shown to be able to produce host beetle pheromone and/or semiochemical compounds (e.g., the beetle antiagreegation hormone, verbenone), particularly in response to host tree chemical compounds [56].
Here, we have identified a series of terpene compounds produced by H. lauricola. How these terpenes may affect the chemical ecology of H. lauricola and its beetel partner within tree galleries, is beyond the scope of this proposal, however, using a sereis of well known bacterial and fungal target species we show signficant antimicrobial activity for several of the compounds. With respect to antimicrobial activity, four of the compounds (2, 4, 7, and 9) exhibited anti-fungal activity against the plant pathogenic fungus, F. oxysporum. Several of the isolated compounds (2, 4-7 towards E. coli and 4-6, 9 towards R. solanacearum) showed antibacterial activity against gram-negative bacteria, however, as can be noted, these sets do not completely overlap suggesting specific antibacterial targets for some of the compounds. Four H. lauricola terpenes (2, 6, 7, and 9) showed antibacterial activity towards gram-positive bacteria (note against B. subtilis but not S. aureus), indicating that some of these compounds are active against both gram-postivie and gram-netative bacteria although with target specific susceptibilty. Fungal diterpened from Sarcodon scabrosus, including compounds sarcodonin L, allocyathin B2, sarcodonin G and sarcodonin L have also been shown to possess antibacterial activities [57]. Terpenoids with the same skeleton often show different biological activities. Via comparisons between the structures of different labdane diterpenoids, a carbonyl group (C-8), a hydroxyl group (C-19) and a lactone ring have been shown to be the main factors affecting antibacterial activity [58, 59], consistent with the structural features and activities of the diterpenoids characterized in this report. In addition, because terpenoids participate in the energy metabolism of mitochondrial intima, they can also indirectly affect the accumulation of energy, including by inhibiting the growth of mycelia and/or producing fungistatic effects [60, 61].
In order to provide a broader dataset for potential bioprospecting, we sought to examine any antiproliferative effects of any of the H. lauricola compounds isolated. Several labdane diterpenoids have previously been shown to be able to inhibit cell proliferation by inducing apoptosis [62, 63]. These effects appear to involve perturbation of mitochondrial membrane potential and increasing intracellular ROS levels. Furthermore, some diterpeniods can cause cell cycle arrest in the G2/M phase at low concentrations and G0/G1 phase arrest at high concentrations [64]. In addition, via enzymatic engineering, the structure of several labdane diterpenoids have been modified to obtain products with enhanced activity [65]. H. lauricola diterpene compounds 4, 5, 6, and 7, all showed antiproliferative activity towards lung, breast, and liver cancer cell lines, with compound 9 showing antiproliferative activity towards a liver cancer cell line, alone. Terpenoids structurally altered can lead to their antitumor activities getting enhanced or diminished [66]. Our data also show that some of the H. lauricola diterpenens show higher activity towards DPPH, which acts as an electron transfer (SET-type), as opposed to superoxide, which is a hydrogen atom transfer (HET-type) free radical [67]. This suggests that the polyhydroxy structure of these terpenoids might have some preferential activity against SET-type radicals. Such scavenging of intracellular reactive oxygen species represents the activity of a direct antioxidant [68, 69]. In combination with the evaluation of anti-tumor activity, we found that several of the isolated H. lauricola derived terpenoids display good inhibitory activity on the proliferation of liver cancer cells also have good antioxidant activity, suggesting a potential relationship between the two activities as the liver is involved in organismal antioxidant process [70, 71]. Studies have also shown that in the oxidative damage model of liver cancer cell etiology, antioxidants can indirectly resist oxidative damage of cells through the expression of antioxidant enzymes and genes, that is, via induction of cellular oxidative stress responses [72]. The terpenoids we obtained may not only scavengers cellular free radicals, but could also be acting to enhance endogenous antioxidant defense systems (antioxidant enzymes and glutathione system), and hence their protective mechanisms against oxidatively damaged cells deserve further investigation. The overall characterization of the H. lauricola diterpenoids and the various activties examined herein suggest that they may play important roles in inhibiting competing microbes (e.g., within the tree gallery), providing resistance against oxidative stress, and even potenially enhancing the nutritive value of the fungus for its beetle host.
Comments 2: No aim, objectives of the study were pronounced.
Response 2: The text has been edited to provide a clear objective in the Introduction section.
…Our objectives were to: (1) identify suitable growth and purification conditions for the isolation of terpenes from H. lauricola, and (2) examine the biopharmaceutical potential of isolated compounds with respect to antibacterial, antifungal, antooxidant, and antproliferative activties.…
Comments 3: In the Methods the fermentation period was not indicated.
Response 3: The fermentation was conducted at 26oC for 28 d in 2.4. in the Method section.
Comments 4: Check medium for large scale fermentation: PDB or RSM.
Response 4: RSM medium was used for large-scale fermentation.
Comments 5: Why the mycelium not culture liquid was used for extraction?
Response 5: We used the mycelia as this would represent a more total analysis of metabolites produced rather than only those that are secreted into the media.
Comments 6: What was a principle of the purification of metabolites?
Response 6: The purification of metabolites followed the solubility parameter close principle.
Comments 7: How authors do explain the choice the organisms for bioassays? The selected ones seem cannot explain chemical ecology of the fungus.
Response 7: Our objectives for the antimicrobial, and antiproliferative assays were to determine a broader and potentially more biopharmaceutically applicable dataset for evaluating the activities of the isolated compounds. As such, we used a standard set of well known prokaryotes, and plant fungal pathogen. As little is known concerning what other microbes would be directly ecologically relevant for this fungus, we did not have significant other possibilities. However, the reviewer makes an excellent point, and we believe that our data could be used in future chemical ecology studies of this fungus.
Comments 8: The title of Figure 1 should be changed. Please, include the fungal name.
Response 8: We have changed the title of Figure 1.
Comments 9: What a characteristic smell of the culture? Please, in more detail.
Response 9: We have edited the text to described the odor as “yeast-like”.
Comments 10: The quality of HPLC chromatograms seems low. The extraction solvent and HPLC conditions were not optimized for the terpenoids.
Response 10: We agree that the conditions could be further optimized for terpenoids, however, our protocols enabled us to purify the compounds satisfactorily.
Comments 11: The mass spectra of compounds 7-9 are of low quality. Please, explain why.
Response 11: Compounds 7, 8 and 9 produced some losses in the process of liquid phase separation.
Comments 12: What organisms are known to produce 1-9? What about of comparison with metabolites known for relative fungi for Harringtonia lauricola.
Response 12: Compounds 1-9 can be produced in the following known organisms as listed in the table. Generally, metabolites isolated from Harringtonia lauricola showed better activity compared with those known for relative fungi.
Compounds |
Organism species |
1 |
Chiloscyphus polyanthus |
2 |
Forsythia suspensa |
3 |
Forsythia suspensa |
4 |
Fruits of Forsythia suspensa |
5 |
Araucaria cunninghamii |
6 |
Abies nukiangensis |
7 |
Drynariafortunei |
8 |
Lichens and fungi |
9 |
Entomopathogenic fungus Hypocrella sp. |
Comments 13: What means “---" in the table 1?
Response 13: “---”means IC50> 100 μM (no anti-tumor effect).
Comments 14: Change Conclusions accordingly to the aims of the manuscript.
Response 14: We have changed Conclusions.
Here, we show that H. lauricola, when cultivatedin the brown rice and Lauraceae species sawdust, produces abundant bioactive compounds, and a total of six labdane diterpenoids and three hopane triterpenes were isolated from fungal cultures. These compounds (1-9) were isolated from H. lauricola, and even more broadly from Ophiostomatales, for the first time. To determine the potential biological and biopharmaceutical function(s) of these substances, all of the compounds were evaluated for antibacterial, antifungal, antiproliferative, and antioxidant bioactivities. Compounds 2, 4 and 6 showed potential antitumor, antibacterial and antioxidant activities. The compound characterized were diterpenoids and the various activties characterized herein suggest that they may play important roles in inhibiting competing microbes (e.g., within the tree gallery), providing resistance against oxidative stress, and even potenially enhancing the nutritive value of the fungus for its beetle host. Our study expands the range of biological activities of these terpenoids, providing a reference for the development and utilization of secondary metabolites, and provides the first clues as to potential contributions of secondary metabolites to the unique lifestyle of this fungus that include the ability to grow as a saprophyte, plant pathogen, and insect (beetle) symbiont. The active compounds described are all small molecular weight terpenoids and aromatic ketones. Our analyses of these compounds indicate the significant structural diversity of active metabolites found in insects, plants and fungi, which can be a rich reserviore for biopharmaceutical discovery and application.
Comments 15: English more or less fine. Some mistakes can be found.
Response 15: We have revised grammatical problems in the paper.
We tried our best to improve the manuscript and made some changes marked in blue in revised paper which will not influence the content and framework of the paper. We appreciate for Editors/Reviewers’ warm work earnestly and hope the revision will meet with your approval. Once again, thank you very much for your comments and suggestions.
Kind regards,
Junzhi Qiu
E-mail address: [email protected]
Round 2
Reviewer 3 Report
Comments and Suggestions for Authors
The authors made an good attempt to improve the manuscript. Some moor problems are left:
1) check the name of compound 1 through text (e.g. p. 10)
2) check Supplimentary
mass spectra presented are not HRESIMS
Fig. S14 seems C-NMR not H-NMR
C-NMR on Fig. S14 seems is not relevant to the compound 5
Author Response
Dear Editors and Reviewers:
Thank you for your letter and comments relating to our manuscript entitled “Characterization of terpenoids from the ambrosia beetle symbiont and laurel wilt pathogen, Harringtonia lauricola” (ID: jof-2720139). The comments were very helpful in revising and improving our manuscript as well as emphasizing the significance to our research. We have read the comments carefully and made corrections accordingly. Revised portions are marked in blue in the manuscript. The main corrections in the paper and our responses to the reviewer’s comments are given below. We hope that the revisions in the manuscript and our accompanying responses will be sufficient to make our manuscript suitable for publication in the Journal of Fungi.
Responses to the comments of the reviewer:
Comments 1: check the name of compound 1 through text (e.g. p. 10).
Response 1: The name of compound 1 has been checked and changed to manool.
Comments 2: check Supplementary.
mass spectra presented are not HRESIMS
Fig. S14 seems C-NMR not H-NMR
C-NMR on Fig. S14 seems is not relevant to the compound 5
Response 2: We regret the error and HRESIMS has been changed to ESI-MS.
Fig.S14 shows us C-NMR.
We have revised C-NMR on Fig.S14.
We tried our best to improve the manuscript and made some changes marked in blue in revised paper which will not influence the content and framework of the paper. We appreciate for Editors/Reviewers’ warm work earnestly and hope the revision will meet with your approval. Once again, thank you very much for your comments and suggestions.
Kind regards,
Junzhi Qiu
E-mail address: [email protected]
